# Inference between Spectra and Structure Across the Periodic Table

## Abstract

X-ray Absorption Spectroscopy (XAS) is a powerful technique for probing local atomic environments, yet its interpretation remains limited by the need for expert-driven analysis, computationally expensive simulations, and element-specific heuristics. Recent advances in machine learning have shown promise for accelerating XAS interpretation, but many existing models are narrowly focused on specific elements, edge types, or spectral regimes. In this work, we present XAStruct, a two pipeline system capable of predicting XAS spectra from crystal structures and inferring local structural descriptors from XAS input. XAStruct is trained on a large-scale dataset spanning over 70 elements across the periodic table, enabling generalization to a wide variety of chemistries and bonding environments. The framework includes the first machine learning approach for predicting neighbor atom types directly from XAS spectra, as well as a generalizable regression model for mean nearest-neighbor distance that requires no element-specific tuning. By combining deep neural networks for complex structure–property mappings with efficient baseline models for simpler tasks, XAStruct offers a scalable and extensible solution for data-driven XAS analysis and local structure inference. The source code will be released upon paper acceptance.

## 1 Introduction

X-ray Absorption Spectroscopy (XAS), which encompasses both the X-ray Absorption Near Edge Structure (XANES) and the Extended X-ray Absorption Fine Structure (EXAFS), is a powerful technique for probing the electronic structure and local chemical environment of atoms (Li et al., 2024; Kerr et al., 2022; Lengeler, 1985; Sharma et al., 2018; Lengeler, 1989). It plays a fundamental role in materials science by enabling a precise evaluation of oxidation states, coordination numbers, and bond lengths, thereby driving progress in areas such as renewable energy, catalysis, and pharmaceuticals. However, acquiring high-quality XAS data remains challenging, especially for rare or sensitive samples like metalloproteins, due to the dependence on synchrotron radiation facilities and complex sample preparation requirements (Dinsley et al., 2022). Additionally, theoretical modeling using conventional approaches such as Density Functional Theory (DFT) is computationally demanding and often inaccurate in systems with strong electronic correlations or complex coordination environments (Chan et al., 2019; Rehr et al., 2010). Analytical techniques for extracting structural descriptors, such as coordination number (CN), mean nearest bond distances (MNND), and neighbor atom types, typically rely on iterative fitting, handcrafted features, or comparison to reference spectra, which are time-consuming, labor-intensive, and difficult to generalize.

To address these limitations, AI-driven approaches have emerged as promising alternatives to accelerate and automate the interpretation of XAS. Inspired by recent breakthroughs such as AlphaFold in protein structure prediction (Senior et al., 2020; Jumper et al., 2021; Abramson et al., 2024) and the broader success of AI in healthcare, astrophysics, and materials discovery (Wang et al., 2024; Shen et al., 2019; Kumar et al., 2023; Levis et al., 2022; Wang et al., 2025), these methods offer the potential for data-driven solutions that scale efficiently and adapt across chemical systems. Existing ML models for the prediction of XAS have focused mainly on crystalline inorganic materials (Kotobi et al., 2023; Gleason et al., 2024; Carbone et al., 2020; Zheng et al., 2020; Torrisi et al., 2020), typically modeling either structure-to-spectra or spectra-to-structure mappings. However, many of these models are limited to a small number of elements, specific absorption edges, or narrow spectrum types, which restricts their generalization and practical utility.

This work presents **XAStruct**, a novel two-stage machine learning framework designed to enhance both the accuracy and completeness of XAS-related predictions. The first stage employs a physics principle-aware graph neural network (GNN) encoder (in this work, the CHGNet (Deng et al., 2023) backbone is utilized) that extracts complex, non-linear features from crystal structures, enabling the accurate prediction of XANES and EXAFS spectra. The second stage goes beyond mere spectral prediction, it includes both classification and regression models to predict not only the type and number of neighboring atoms around the absorbing atom, such as Oxygen (O), Sulfur (S), and Copper (Cu), but also the mean nearest neighbor distance. As shown in Supplementary Figure S14, XAStruct is trained on a large-scale dataset spanning **over 70 elements across the periodic table**, covering both K- and L-edge XANES and EXAFS regimes, which enables broad generalization to diverse chemistries.

In particular, the XAStruct proposed in this work offers the following key contributions:

- Enables learning in both structure-to-spectra and spectra-to-structure pipelines, within a two pipeline framework trained across a wide range of absorbing atoms.

- Achieves periodic table wide coverage, enabled by training on over 70 elements, rather than focusing on a single element or material class.

- Provides the first ML-based prediction of neighbor atom types directly from XAS spectra, offering interpretable insights into local atomic environments and addressing a long-standing inverse problem in XAS analysis.

- Offers a generalizable model for MNND prediction from spectra, serving as a robust surrogate for geometric interpretation and structure refinement.

## 2 RELATED WORKS

**GNN-Based Models for XAS Prediction**  Recent machine learning efforts have explored graph neural networks (GNNs) to predict XAS spectra from atomic structures (Kotobi et al., 2023; Gleason et al., 2024; Carbone et al., 2020). However, most existing approaches are limited in scope, focusing on multilayer perceptrons applied to graph embeddings and often restricted to a single element (e.g., Cu) or edge type (e.g., L-edge) Gleason et al. (2024). Many models avoid the more complex K-edge spectra, which are more sensitive to coordination environments and geometric oscillations, and tend to focus instead on light elements (e.g., O and N) in small molecules (Kotobi et al., 2023; Carbone et al., 2020), neglecting the broader chemical diversity. Furthermore, the closed-source nature of several of these models limits reproducibility and wider adoption.

In contrast, our work introduces a learning framework built around a GNN-based architecture designed for learning across **over 70 elements**, and capable of predicting both **K-edge and L-edge XANES and EXAFS spectra**. By explicitly encoding elemental identity through masking and leveraging architectural components such as GatedLinear layers and SwiGLU activations, our model achieves strong generalization across edge types, energy regions, and diverse chemical environments. This approach marks a significant step beyond prior work focused on isolated systems, offering a scalable and interpretable framework for comprehensive XAS prediction.

**Structure Property Prediction from XAS**  Machine learning approaches for extracting structural information from XAS spectra have largely focused on predicting descriptors such as CN, MNND, or oxidation state using classification or regression models trained on simulated data (Carbone et al., 2019; Torrisi et al., 2020; Zhan et al., 2025; Martini et al., 2020; Guda et al., 2021; Zheng et al., 2020; Narong et al., 2024; Smolentsev & Soldatov, 2007). Most existing methods rely on handcrafted spectral features or fixed featurization pipelines and always require separate models per element, limiting scalability across chemically diverse systems. Random forest-based methods (Torrisi et al., 2020; Zheng et al., 2020; Narong et al., 2024) have been applied to CN and MNND prediction in limited chemical spaces (e.g., transition metal oxides), while CNN-based models like Carbone et al. (Carbone et al., 2019) classify local chemical environments but are constrained to specific elements. Although recent GNN-based models (Zhan et al., 2025) have begun to address structure-from-spectrum learning, they still lack support for multi-element generalization and multi-target prediction.

In contrast to the above work, ours introduces the first generalizable MNND prediction model capable of generalizing across more than 70 elements using shared weights. And to our knowledge, this work is the first to explore the prediction of neighbor atom types directly from XAS spectra. While CN and atom-type models are still trained per element due to categorical variance, our approach avoids reliance on handcrafted features and demonstrates that structural descriptors can be learned in a scalable, extensible manner, balancing interpretability, generalization, and precision in spectrum-based structure prediction.

## 3 THEORETICAL ANALYSIS

In this section, we first present the physical background of XAS and its connection to local structural properties, highlighting the absence of accurate analytical expressions for XAS and the challenges it raises for modeling. We then define the structural descriptors used as prediction targets in our study, and explain their physical significance and spectral relevance. Finally, we formulate the supervised learning tasks of two complementary tasks, detailing both the forward task of predicting spectra from structure and the inverse task of predicting structural properties from spectra.

### 3.1 X-RAY ABSORPTION SPECTROSCOPY OVERVIEW

X-ray Absorption Spectroscopy (XAS) is a powerful technique for probing the local electronic and atomic structure of materials. Based on the Beer–Lambert law (Bouguer, 1729), the attenuation of X-ray intensity is governed by the linear absorption coefficient $\mu(E)$, which depends on photon energy $E$, sample thickness $d$, and material composition (Timoshenko & Roldan Cuenya, 2020; Newville, 2014). For monatomic systems, an empirical relation approximates $\mu(E) \propto \rho Z^4/(AE^3)$, where $Z$ and $A$ are atomic number and mass, respectively.

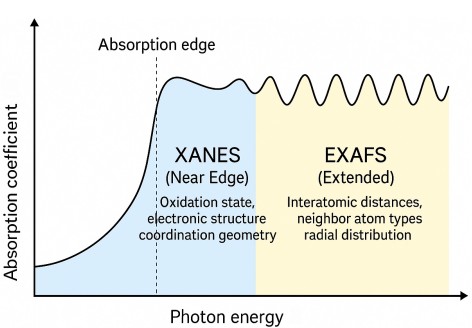

Figure 1: X-ray absorption spectrum with XANES (near-edge) and EXAFS (oscillatory) regions.

A sharp increase in $\mu(E)$, known as the *absorption edge*, appears when the X-ray energy matches the binding energy of a core electron, typically categorized as K-edges or L-edges. K-edges probe geometry at higher energies, while L-edges capture valence and spin-orbit effects. These element-specific edges encode rich structural signals. As shown in Figure 1, XAFS is generally divided into:

- **XANES (X-ray Absorption Near Edge Structure)**: Reflects electronic structure and coordination geometry. Lacks closed-form modeling and is typically resolved using expensive quantum simulations.
- **EXAFS (Extended X-ray Absorption Fine Structure)**: Encodes interatomic distances, neighbor types, and coordination. More analytically accessible using scattering path expansions.

While EXAFS is physically more tractable, XANES encodes richer chemical information. The absence of closed-form expressions for XANES, along with its non-trivial sensitivity to structural and electronic configurations, makes it particularly well-suited for machine learning. Deep learning models can bridge this gap by learning structure-property relationships directly from spectra, without relying on handcrafted physical approximations. These characteristics form the foundation for the supervised learning tasks addressed in this work, including XAFS prediction from structure and inverse prediction of coordination number, mean nearest-neighbor distance, and neighbor atom types from spectra.

### 3.2 STRUCTURAL DESCRIPTORS: DEFINITIONS AND SIGNIFICANCE

To characterize the local environment of absorbing atoms, we focus on three widely used structural descriptors: CN (coordination number), MNND (mean nearest neighbor distance), and nearest-

neighbor atom types. These descriptors are fundamental to conventional XAS analysis and directly influence XAS spectral features. MNND, treated as a continuous regression target, reflects the average bond length around the absorbing atom and governs the frequency and phase of EXAFS oscillations. CN, which can be predicted as a classification task, captures the number of nearby atoms and is closely tied to structural motifs and absorbing intensities in XANES. The identities of nearest-neighbor atoms which are modeled as categorical outputs, define the chemical environment and significantly affect spectral shape via changes in scattering and electronic structure. Together, these descriptors provide a chemically meaningful, physically grounded foundation for training and evaluating machine learning models in spectrum-structure analysis.

### 3.3 PROBLEM FORMULATION AS LEARNING TASKS

Let a material structure be represented by a graph $\mathcal{G} = (\mathcal{V}, \mathcal{E})$, where $\mathcal{V}$ is the set of atoms (nodes) and $\mathcal{E}$ is the set of interatomic interactions (edges), typically constructed using geometric or chemical heuristics (e.g., distance cutoff or coordination rules). Each node $v \in \mathcal{V}$ is associated with a feature vector $\mathbf{h}_v$ encoding elemental and structural properties. Let $\mathbf{E} = [E_1, \ldots, E_n] \in \mathbb{R}^n$ denote the sampled energy axis of the X-ray absorption spectrum, and $\mathbf{x} = [\mu(E_1), \ldots, \mu(E_n)] \in \mathbb{R}^n$ the corresponding absorption coefficients.

We consider two complementary learning tasks: (1) predicting XAS spectra from crystal structures, and (2) recovering structural descriptors from spectra. These tasks span two linked domains of structure and spectroscopy, but are modeled with independently trained systems.

**Forward Task (Structure → Spectrum)** Given a structure graph $G$, absorbing element $z \in \mathbb{Z}^+$, and energy axis $\mathbf{E} \in \mathbb{R}^n$, the goal is to predict the absorption spectrum $\mathbf{x} \in \mathbb{R}^n$. This task is formulated as:

$$f_\theta : (G, z, \mathbf{E}) \to \hat{\mathbf{x}} \tag{1}$$

where $f_\theta$ is a neural model with parameters $\theta$, producing $\hat{\mathbf{x}}$ as the predicted absorption signal.

**Inverse Task (Spectrum → Structure)** Given an input spectrum $\mathbf{x} \in \mathbb{R}^n$ and absorbing element $z \in \mathbb{Z}^+$, we aim to infer key local structural descriptors:

$$g_\phi : (\mathbf{x}, z) \to (\hat{d}, \hat{c}, \hat{t}) \tag{2}$$

Here, $\hat{d} \in \mathbb{R}$ is the predicted mean nearest neighbor distance (MNND), $\hat{c} \in \mathbb{Z}^+$ the coordination number (CN), and $\hat{t} \in \mathbb{Z}^+$ the nearest neighbor atom type. The function $g_\phi$ is parameterized by weights $\phi$ and serves to recover geometric information from spectral patterns.

These two tasks reflect the dual nature of XAS analysis—simulation and interpretation—and form the basis for our model design. While trained separately, both mappings are essential to enabling data-driven reasoning about structure-spectrum relationships across a wide chemical and spectral space.

## 4 XASTRUCT: BRIDGING XAS AND STRUCTURAL INFERENCE

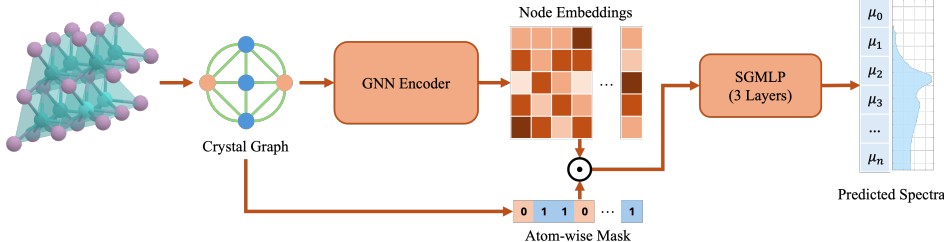

Figure 2: Structure to Spectrum: Predicting XANES/EXAFS from Crystal Graph

Figure 3: Spectrum to Structure: Predicting CN/MNND/Neighbor Atom from XAFS

While powerful, traditional XAS analysis is often limited by computationally intensive simulations and the ill-posed nature of inverting spectra to find structure. Existing machine learning models often lack generalizability, being restricted to specific elements or spectral regions.

To address these challenges, we propose **XAStruct**—short for Bridging **XAS** and **Struct**ural Inference. Although the structure-to-spectrum and spectrum-to-structure tasks are trained independently, they represent two complementary directions of XAFS-driven learning in crystalline materials. As illustrated in Figure 2,3, XAStruct is composed of two distinct components: a forward pipeline that predicts XANES and EXAFS spectra from crystal graphs, and an inverse pipeline that infers structural descriptors directly from XAFS spectra. In the sections that follow, we describe each component in detail.

**Structure-to-Spectrum**    The forward model $f_\theta$ accepts a local atomic structure as a graph $G = (V, E)$, where $V$ denotes atoms and $E$ denotes interatomic interactions determined by a spatial cutoff (6 angstroms in our tasks). Each node $v_i \in V$ is initialized with a feature vector $\mathbf{h}_i^{(0)} \in \mathbb{R}^{d_{\text{in}}}$ that encodes atomic identity and geometric information.

The graph is processed by a physics principle-aware GNN encoder $F_{\text{gnn}}$, in this work, which is based on CHGNet Deng et al. (2023), produces node embeddings:

$$\mathbf{H} = F_{\text{gnn}}(G) \in \mathbb{R}^{|V| \times d} \tag{3}$$

To isolate the local environment of the absorbing atom, we apply a binary mask vector $\mathbf{m} \in \{0,1\}^{|V|}$ indicating the absorber and its immediate neighbors. The masked mean embedding is computed as:

$$\mathbf{z}_{\text{struct}} = \frac{1}{|V|} \sum_{i=1}^{|V|} m_i \cdot \mathbf{H}_i \in \mathbb{R}^d \tag{4}$$

This structural embedding is passed through a 3-layer (2 blocks) **S**wiGLU-**G**ated **M**ulti-**L**ayer **P**erceptron (**SGMLP**) $F_{\text{out}}$ to generate the predicted X-ray absorption spectrum:

$$\hat{\mathbf{x}} = f_\theta(G, z, \mathbf{E}) = F_{\text{out}}(\mathbf{z}_{\text{struct}}) \in \mathbb{R}^n \tag{5}$$

Each block of $F_{\text{out}}$ consists of the following components:

- **GatedLinear(GL)**: a linear projection with a gating branch, for input $\mathbf{x} \in \mathbb{R}^d$:

$$\texttt{GL}(\mathbf{x}) = (\mathbf{W}_v \mathbf{x} + \mathbf{b}_v) \cdot \sigma(\mathbf{W}_g \mathbf{x} + \mathbf{b}_g)$$

where $\mathbf{W}_v, \mathbf{W}_g \in \mathbb{R}^{d_{\text{out}} \times d}$ are learned weights, $\mathbf{b}_v, \mathbf{b}_g \in \mathbb{R}^{d_{\text{out}}}$ are learned biases and $\sigma$ is the Sigmoid activation.

- **SwiGLU**: a parameterized activation function, for input $\mathbf{x} \in \mathbb{R}^d$:

$$\texttt{SwiGLU}(\mathbf{x}) = \left( \frac{\mathbf{x}\mathbf{W}_g + \mathbf{b}_g}{1 + e^{-\beta(\mathbf{x}\mathbf{W}_g + \mathbf{b}_g)}} \right) \otimes (\mathbf{x}\mathbf{W}_v + \mathbf{b}_v)$$

  where $\mathbf{W}_v, \mathbf{W}_g \in \mathbb{R}^{d_{\text{out}} \times d}$ are learned weights, $\mathbf{b}_v, \mathbf{b}_g \in \mathbb{R}^{d_{\text{out}}}$ are learned biases and $\beta \in \mathbb{R}$ is a learned parameter.

- **LayerNorm**: applied for stabilizing training.

The full SGMLP Block is composed of:

$$\texttt{SBlock}(\mathbf{x}) = \texttt{SwiGLU}(\texttt{LayerNorm}(\texttt{GL}(\mathbf{x})))$$

And for a $k$-layer ($k-1$ blocks) SGMLP, it is constructed in this way:

$$\texttt{SGMLP}(\mathbf{x}) = \texttt{GL}\left( \texttt{SBlock}^{(k-1)} \circ \cdots \circ \texttt{SBlock}^{(1)}(\mathbf{x}) \right) \tag{6}$$

**Spectrum-to-Structure**   For the inverse mapping $g_\phi$, we begin with an input spectrum $\mathbf{x} \in \mathbb{R}^n$ and energy axis $\mathbf{e} \in \mathbb{R}^n$. The prediction of CN is based on the random forest model $f_c$ Breiman (2001):

$$\hat{c} = \arg\max f_c([\mathbf{x}_{\text{xanes}} \| \mathbf{x}_{\text{exafs}}]), \quad f_c(\mathbf{x}) \in \mathbb{R}^C \tag{7}$$

As for MNND and neighbor atom type predictions, depending on the spectrum type (e.g., XANES and EXAFS), we define two parallel embedding blocks: $f_{\text{E,xanes}}, f_{\text{A,xanes}}$ for XANES energy and absorption; $f_{\text{E,exafs}}, f_{\text{A,exafs}}$ for EXAFS inputs. Each embedding block is an SGMLP composed of the same gated architecture as described above.

Let $\mathbf{z}_{\text{e}}$ and $\mathbf{z}_{\text{a}}$ be the embedded energy and absorption vectors:

$$\mathbf{z}_{\text{e},*} = f_{\text{E},*}(\mathbf{e}), \quad \mathbf{z}_{\text{a},*} = f_{\text{A},*}(\mathbf{x}) \tag{8}$$

where $*$ indicates either XANES or EXAFS depending on the context.

For predicting the neighbor atom types, the concatenated $\mathbf{z}_{\mathbf{a},*}$ latent vectors are fed into SGMLP $f_t$, yielding:

$$\hat{t} = \arg\max f_t([\mathbf{z}_{\mathbf{a},\text{xanes}} \| \mathbf{z}_{\mathbf{a},\text{exafs}}]), \quad f_t(\mathbf{x}) \in \mathbb{Z}^{+m} \tag{9}$$

For predicting MNND, the latent vectors $\mathbf{z}_{\text{e},*}$ and $\mathbf{z}_{\text{a},*}$ are concatenated into joint latent vectors:

$$\mathbf{z}_{\mathbf{x},*} = [\mathbf{z}_{\text{e},*} \| \mathbf{z}_{\text{a},*}] \in \mathbb{R}^{2d} \tag{10}$$

then they are passed through corresponding convolution blocks $f_{\text{conv},*}$, composed of convolution and pooling layers:

$$f_{\text{conv}}(\mathbf{z}_{\mathbf{x},*}) = \texttt{AvgPool}(\texttt{ReLU}(\texttt{BN}(\texttt{Conv1D}(\mathbf{z}_{\mathbf{x},*})))) \tag{11}$$

After that, the final features of XANES and EXAFS are concatenated and fed into SGMLP $f_d$, yielding:

$$\hat{d} = f_d([\mathbf{z}_{\mathbf{x},\text{xanes}} \| \mathbf{z}_{\mathbf{x},\text{exafs}}]) \in \mathbb{R} \tag{12}$$

## 5   EXPERIMENT

While prior datasets Ewels et al. (2016); Chen et al. (2021) have been used for XAS prediction, they do not contain sufficient input information (e.g., full 3D structure graphs or energy-resolved grids) required for evaluating our model. Therefore, we collected consistent structure-spectrum pairs across 70+ elements (see Figure **??** and Figure S14 for details), from Materials ProjectJain et al. (2013), which supports both forward and inverse learning tasks. Resulting in a dataset comprising over 43,000 unique crystal structures and approximately 120,000 corresponding X-ray absorption spectra (XANES and EXAFS), spanning both K and L edges across more than 70 chemical elements. The datasets are split into 8:2 subsets for training and validation. We conduct extensive experiments on the constructed dataset and train some models for each task under tailored settings to accommodate the complexity and variability inherent in different spectral and structural properties. The experiments are run on four NVIDIA RTX A6000-48GB GPUs.

## 5.1 Predicting K/L-edge XANES and EXAFS from Crystal Graphs

This experiment assesses the structure-to-spectrum capabilities of our proposed framework by predicting K- and L-edge XANES, as well as K-edge EXAFS (L-edge EXAFS dataset is not available), directly from crystal graphs. Each material is encoded as a structure graph with node and edge features that capture atomic identities and geometric relationships. The GNN encoder, followed by an SGMLP head, maps local atomic environments to energy-dependent absorption spectra.

To account for the sensitivity of spectral features to the absorbing element and edge type, we train separate models for each element–edge–spectrum combination. This element-specific training avoids cross-element feature drift and allows models to specialize in chemically consistent environments. Models are trained using Mean Squared Error (MSELoss) between predicted and reference spectra, and optimized with AdamW (learning rate $10^{-4}$, weight decay 0.01).

We benchmark XAStruct against three common GNN baselines (GCNKipf & Welling (2016), GATVeličković et al. (2017), and MPNNGilmer et al. (2017)) and CuXASNet Gleason et al. (2024), which is specialized for copper L-edge XANES. For CuXASNet, we report the original paper's numbers due to the lack of open-source code; for others, we reimplement and train them under the same data split.

Table 1 and Figure S3, S2, S4 in the supplementary material present the quantitative results, showing that our model remarkably outperforms general-purpose GNNs and achieves much lower MAE on both global and edge-specific evaluations. Particularly, on the Cu L-edge benchmark, XAStruct achieves an MAE of 0.0012, substantially improving over the 0.0391 reported by CuXASNet. These results highlight XAStruct's ability to learn precise local electronic structures from geometry alone. Furthermore, compared to other general-purpose GNN models, XAStruct achieves a lower mean absolute error (MAE) of 0.0537/0.0031 for XANES K-edges/L-edges and 0.0302 for EXAFS(K-edge) prediction, highlighting its strong adaptability across diverse elements and spectral regimes.

Table 1: MAE comparison for XANES and EXAFS prediction across models. XANES results are reported on K-edge, L-edge, and Cu L-edge subsets; EXAFS is evaluated only on K-edge spectra (L-edge data unavailable).

| Model | XANES (K) | XANES (L) | XANES (Cu, L) | EXAFS (K) |
|---|---|---|---|---|
| MPNN | $0.1026_{\pm 0.0004}$ | $0.0825_{\pm 0.0008}$ | $0.0807_{\pm 0.0007}$ | $0.0931_{\pm 0.0002}$ |
| GCN | $0.0997_{\pm 0.0005}$ | $0.0837_{\pm 0.0009}$ | $0.0898_{\pm 0.0006}$ | $0.1044_{\pm 0.0001}$ |
| GAT | $0.1005_{\pm 0.0004}$ | $0.0718_{\pm 0.0008}$ | $0.0844_{\pm 0.0006}$ | $0.0901_{\pm 0.0001}$ |
| CuXASNet | – | – | $0.0391_{\pm 0.0007}$ | – |
| **XAStruct (Ours)** | $\mathbf{0.0537}_{\pm 0.0005}$ | $\mathbf{0.0031}_{\pm 0.0008}$ | $\mathbf{0.0012}_{\pm 0.0006}$ | $\mathbf{0.0302}_{\pm 0.0001}$ |

## 5.2 Predicting Structural Descriptors from K-edge XAFS Spectra

In this section, we investigate the ability of machine learning models to infer local structural descriptors of the absorbing atom directly from its corresponding K-edge XAFS spectrum (including XANES and EXAFS), as L-edge data is excluded due to missing L-edge EXAFS in the dataset. These descriptors include MNND, CN, and the categorical identities of nearest neighbor atoms. Each of these tasks provides complementary insight into local bonding environments and coordination chemistry, and presents unique challenges in terms of learning and generalization.

**Generalizable Model for MNND Prediction** For MNND prediction, we propose a single, generalizable SGMLP-Convolution hybrid regression model that accepts the full XAFS spectrum (including $\mu(E_i)$ and $E_i$, as shown in Fig 3) as input. The model is trained across all elements jointly, enabling it to generalize across more than 70 elements, including transition metals, main-group elements, and rare earths. Despite the chemical diversity in the training set, this model achieves robust performance without element-specific tuning, demonstrating remarkable generalizability when compared to other baseline models trained on element-specific datasets.

**CN and Neighbor Atom Classification** In contrast to MNND, predicting CN and neighbor atom types raises a more discrete and element-specific classification challenge. CN values typically range

Table 2: Comparison of structure descriptor prediction performance between XAStruct and baseline models.

| Task | Metric | XAStruct (Ours) | Baseline | Baseline Model |
|---|---|---|---|---|
| MNND | MAE | $\mathbf{0.0350}_{\pm 0.0007}$ | $0.0557_{\pm 0.0009}$ | Random Forest |
| CN | F-1 Score | $61.35_{\pm 0.01}\%$ | $\mathbf{61.44}_{\pm 0.02}\%$ | MLP |
| | Accuracy | $\mathbf{69.26}_{\pm 0.02}\%$ | $68.65_{\pm 0.03}\%$ | |
| Neighbor Atom | F-1 Score | $\mathbf{88.76}_{\pm 0.01}\%$ | $82.23_{\pm 0.01}\%$ | Random Forest |
| | Accuracy | $\mathbf{92.96}_{\pm 0.02}\%$ | $88.99_{\pm 0.01}\%$ | |

over a small set (e.g., 2, 3, 4, 5, 6, 8, 12), but vary significantly across different elements and crystal classes. Neighbor atom classification is even more difficult due to the high variance (over 70 element types) and sparsity of classes in any given crystal structure. For example, while there are 70+ possible atom types in the full dataset, any specific XAFS spectrum is typically associated with only 2–4 neighboring species, resulting in highly imbalanced and underrepresented labels.

Given this complexity, we adopt two different approaches: for CN prediction, we use random forest classifiers trained independently for each element. Attempts to train models on the full dataset failed due to poor convergence and significant performance degradation, particularly for underrepresented CN values. For neighbor atom prediction, we use our proposed SGMLP-based deep classifier, also trained separately per element. This choice is motivated by the extreme label imbalance and high inter-class variance, which make unified classification across all elements nearly impractical.

**Baseline Justification and Training Setup**  Although deep learning architectures such as CNNs and GNNs have been explored for structure prediction in related domains, prior studies have consistently demonstrated that random forest models outperform these approaches for structure descriptor prediction from XAS spectra Zheng et al. (2020); Torrisi et al. (2020); Carbone et al. (2019); Zhan et al. (2025). Therefore, we adopt random forest as our primary baseline, alongside standard MLP classifiers, and compare against deep neural models only where appropriate. Specifically, neighbor atom and CN classification models are trained using the CrossEntropyLoss, while the MNND regression task uses Mean Squared Error Loss (MSELoss). All deep models are optimized using AdamW with a learning rate of $10^{-4}$ and a weight decay of 0.01.

As shown in Table 2 and Figures S10, S12, and S8 (supplementary), XAStruct achieves strong performance across all structure-related prediction tasks. The generalizable MNND model generalizes well across elements and bond distances, capturing the underlying geometric trends with high fidelity. In contrast, as illustrated in Figure S9, the random forest baseline exhibits systematic errors—overestimating MNND for tightly packed structures and underestimating it for loosely coordinated environments, resulting in a lower regression slope (0.92) compared to XAStruct (0.97). For neighbor atom classification, XAStruct achieves notably higher precision and recall, especially in rare or chemically diverse cases, highlighting its advantage over conventional models. While the random forest outperforms MLP in CN prediction, its lightweight nature also yields faster training and inference, making it a practical baseline. Overall, these results validate our design choices and emphasize the benefits of combining task-specific architectures with robust learning objectives.

## 5.3 ABLATION STUDY ON ARCHITECTURAL COMPONENTS

To validate the design choices of our architecture, we conducted an ablation study focusing on the key components of our SGMLP module, which is central to both the forward and inverse pipelines. The study aims to quantify the performance impact of the GatedLinear layer and the SwiGLU activation function. We evaluated four model configurations on the most representative tasks: K-edge XANES prediction, MNND prediction, and neighbor atom classification.

The configurations are as follows:

- XAStruct (Ours): The full proposed model utilizing the complete SGMLP block (Gated-Linear layers with SwiGLU activation).

- SwiGLU: A variant where the SwiGLU activation in the SGMLP is replaced with a standard ReLU activation, while keeping the GatedLinear layer.
- GatedLinear: A variant where the GatedLinear layer is replaced with a standard nn.Linear layer, while keeping the SwiGLU activation.
- Standard MLP: A baseline model where the entire SGMLP block is replaced by a conventional Multi-Layer Perceptron composed of standard nn.Linear layers and ReLU activation functions.

Table 3: Ablation study of XAStruct vs. variants with key components removed. Averages are over the validation set.

| Model Configuration | Structure → Spectrum | Spectrum → Structure | | |
| --- | --- | --- | --- | --- |
| | XANES (K) MAE | MNND MAE | Neighbor. F1 (%) | Neighbor. Acc. (%) |
| **XAStruct (Ours)** | **$0.0537_{\pm 0.0005}$** | **$0.0350_{\pm 0.0007}$** | **$88.76_{\pm 0.01}$** | **$92.96_{\pm 0.02}$** |
| SwiGLU (use ReLU) | $0.0612_{\pm 0.0006}$ | $0.0421_{\pm 0.0005}$ | $85.15_{\pm 0.01}$ | $90.12_{\pm 0.01}$ |
| GatedLinear (use Linear) | $0.0595_{\pm 0.0007}$ | $0.0403_{\pm 0.0005}$ | $86.32_{\pm 0.01}$ | $91.05_{\pm 0.02}$ |
| Standard MLP | $0.0689_{\pm 0.0008}$ | $0.0498_{\pm 0.0007}$ | $83.45_{\pm 0.01}$ | $88.54_{\pm 0.03}$ |

The results, summarized in Table 3, clearly demonstrate the benefits of our chosen components. In all evaluated tasks, the full XAStruct model achieves the best performance. Removing either the SwiGLU activation or the GatedLinear layer leads to a noticeable degradation in performance. Replacing the GatedLinear layer with a standard linear layer while keeping SwiGLU (- GatedLinear) results in a moderate drop in accuracy and F1-score, suggesting that the gating mechanism is effective at controlling information flow.

More significantly, replacing SwiGLU with ReLU causes a more substantial performance decrease across all metrics. This highlights the advantage of SwiGLU's non-linear, data-dependent activation for capturing the complex relationships in spectral data. Finally, the standard MLP baseline shows the poorest performance, confirming that the combination of both gating and the advanced activation function in our SGMLP architecture is crucial for achieving state-of-the-art results.

## 6    LIMITATIONS AND CONCLUSION

This work presents XAStruct, a machine learning framework for interpreting and predicting XAS with broad element coverage. Trained on datasets spanning over 70 elements, XAStruct supports both structure-to-spectrum and spectrum-to-structure tasks, covering K-edge and L-edge XANES and EXAFS regions. The key contributions include: (1) a two-pipeline architecture that enables complementary predictions from both structural and spectroscopic inputs; (2) the first machine learning model to predict neighbor atom types directly from XAS spectra—addressing a long-standing inverse problem in spectral interpretation; (3) a generalizable, element-aware model for predicting MNND without element-specific tuning; and (4) a principled approach to structural descriptor prediction, using lightweight random forests for CN and deeper SGMLP-based networks for more complex tasks like MNND and neighbor atom classification. Together, these components make XAStruct a scalable, interpretable, and broadly applicable framework for XAS analysis and local structure inference.

However, several limitations remain. While XAStruct supports dual learning objectives, the two pipelines are trained independently. Additionally, CN and neighbor atom prediction models must be trained separately per element, as unified models struggle with class imbalance and high categorical variance. Our structural predictions still require expert interpretation to reconstruct full geometries, limiting full automation. Moreover, although the MNND model generalizes well, CN and neighbor atom predictions are bounded by closed-set assumptions and limited resolution in fine spectral features such as XANES pre-edges and EXAFS oscillations. These challenges highlight promising directions for future work, including cross-modal alignment, few-shot generalization for underrepresented chemistries, and physically grounded model regularization to enhance robustness and interpretability.

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

# A APPENDIX

## A.1 XAFS THEORETICAL BACKGROUND

Figure S1 presents a typical Cu K-edge X-ray absorption spectrum, illustrating the fundamental decomposition of the total absorption signal into the near-edge (XANES) and extended (EXAFS) regions. The sharp increase in absorption around 8980–8990 eV corresponds to the ionization of a Cu 1s core electron and marks the absorption edge. The edge jump $\Delta\mu_{Cu}$ quantifies the magnitude of this transition and is used for normalization in EXAFS analysis.

To the left of the edge, a weak but resolvable pre-edge feature appears, often associated with 1s $\rightarrow$ 3d or 4p transitions facilitated by hybridization or local symmetry breaking. The red inset illustrates this as a local Cu–O cluster. To the right of the edge, the XANES region spans roughly 30–50 eV and encodes multiple-scattering, oxidation state, and coordination effects. The EXAFS region follows, showing oscillations in $\mu(E)$ caused by interference between the outgoing photoelectron wave and waves backscattered by neighboring atoms. These oscillations decay with energy and distance due to the finite photoelectron mean free path.

The right-side inset schematically illustrates this EXAFS process: the central absorbing atom (Cu) emits a photoelectron wave that is partially backscattered by neighboring atoms (Cu, O), producing interference that modulates the absorption. The resulting spectrum thus encodes both electronic (XANES) and geometric (EXAFS) information, motivating the development of data-driven

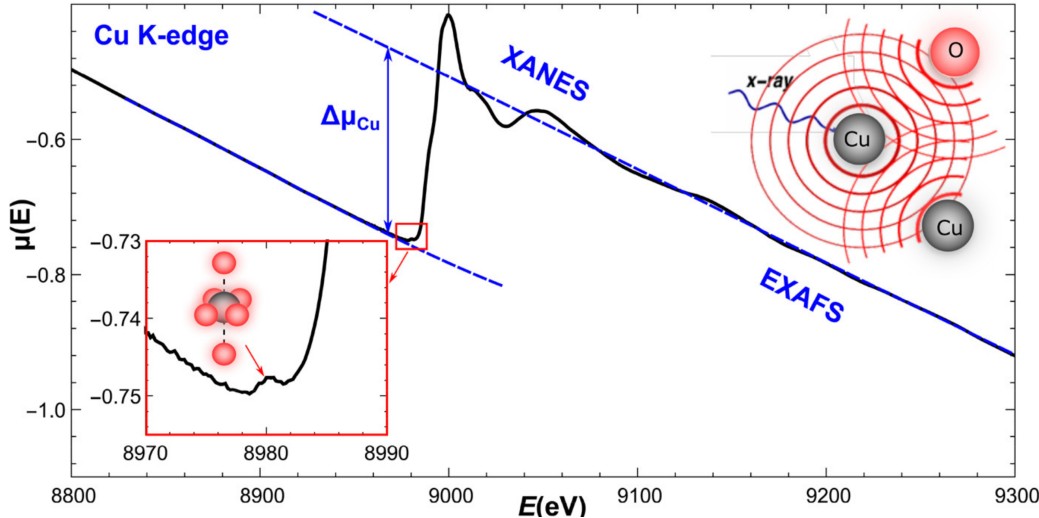

Figure S1: Example of the XAFS spectrum(Timoshenko & Roldan Cuenya, 2020).

approaches (shared-weights where feasible, per-element where necessary) to infer structure and property from such rich spectral data.

The subsequent sections provide more detailed theoretical backgrounds for both XANES and EXAFS interpretation.

**XANES: X-ray Absorption Near Edge Structure**  XANES arises from photoelectron transitions into unoccupied bound states and contains rich information about the electronic structure of the absorbing atom. Governed by Fermi's Golden Rule, the absorption coefficient is expressed asTimoshenko & Roldan Cuenya (2020); Newville (2014); Bouguer (1729):

$$\mu(E) \approx \sum_f \left| \langle f|\hat{T}|i\rangle \right|^2 \delta(\epsilon_f - \epsilon_i - E) \tag{13}$$

where $|i\rangle$ and $|f\rangle$ denote the initial and final states, and $\hat{T}$ is the transition operator. In practice, dipole or quadrupole approximations are applied depending on the edge and transition type.

The position of the absorption edge typically shifts to higher energies with increasing oxidation state, offering a nearly linear correlation in many systems (e.g., Co, Ni, Cu). However, this "edge position" lacks a universally accepted definition and may also reflect coordination charge, making quantitative extraction of oxidation state nontrivial.

Beyond the edge position, several spectral features provide deeper insights:

- **Pre-edge features**, arising from transitions to localized unoccupied states (e.g., 3d), are sensitive to oxidation state and site symmetry. For example, Fe K-edge pre-edge features distinguish Fe(II) from Fe(III), and Cu(II) exhibits a pre-edge at ∼8980 eV absent in Cu(I) due to the fully filled 3d shellTimoshenko & Roldan Cuenya (2020).

- **White lines (WL)**, observed prominently at L-edges of transition metals, result from transitions to unoccupied d-states. The WL intensity correlates with the density of final states and is influenced by oxidation, bonding, and ligand field effects. In K-edge spectra, intense WLs are often absent unless hybridization enables dipole-allowed transitionsTimoshenko & Roldan Cuenya (2020).

These features are typically modeled by fitting Lorentzian or Voigt profiles to experimental spectra, or by integrating areas under peaks after subtracting arctangent baselinesTimoshenko & Roldan Cuenya (2020). Such procedures are necessary because instrumental broadening and core-hole lifetime effects distort line shapes.

Due to its sensitivity and low susceptibility to thermal disorder, XANES is ideal for in situ or operando studies. However, the lack of a tractable analytical model necessitates using empirical fingerprinting or first-principles simulation (e.g., FEFFNewville (2001a;b), FDMNESBunău et al. (2024)) to support interpretation.

**EXAFS: Extended X-ray Absorption Fine Structure**   EXAFS originates from the interference between the outgoing photoelectron and waves scattered by neighboring atoms, leading to oscillations superimposed on the absorption coefficient $\mu(E)$. The EXAFS contribution $\chi(k)$ is typically modeled as:

$$\chi(k) = \sum_p \chi_p(k) \tag{14}$$

with each term approximated using a single-scattering model:

$$\chi_p(k) = \frac{S_0^2}{kR^2} \int_{R_i}^{R_j} g_p(R) F_p(k, R) e^{-2R/\lambda(k)} \sin\left(2kR + \phi_p(k, R)\right) dR \tag{15}$$

where $g_p(R)$ is the partial radial distribution function, $F_p(k, R)$ is the backscattering amplitude, and $\phi_p(k, R)$ is the phase shift. $k = \sqrt{2m_e(E - E_0)/\hbar^2}$ is the photoelectron wavenumber.

The EXAFS region primarily probes short-range order and is relatively insensitive to electronic structure. Coordination number, interatomic distances, and disorder parameters can be extracted through curve fitting to theoretical models (e.g., FEFFNewville (2001a;b)). However, disorder and thermal vibrations limit the interpretability of higher coordination shells.

Though less chemically sensitive than XANES, EXAFS enables semi-quantitative structure determination, and the shift in the fitted $E_0$ parameter across conditions may indirectly reflect oxidation state changes. Nevertheless, EXAFS is best interpreted in combination with XANES to capture both electronic and geometric structure.

A.2   METRICS USED IN EXPERIMENTS

To quantitatively assess the performance of XAStruct across prediction tasks, we employ the following standard metrics:

**Mean Absolute Error (MAE):**   Used for regression tasks such as MNND and spectral prediction, MAE measures the average magnitude of absolute errors:

$$\text{MAE} = \frac{1}{N} \sum_{i=1}^{N} |y_i - \hat{y}_i| \tag{16}$$

where $y_i$ and $\hat{y}_i$ are the ground truth and predicted values, these notations are used consistently in the following paragraphs within this section.

**Coefficient of Determination ($R^2$):**   Evaluates how well the predicted values approximate the true values in regression:

$$R^2 = 1 - \frac{\sum_{i=1}^{N}(y_i - \hat{y}_i)^2}{\sum_{i=1}^{N}(y_i - \bar{y})^2} \tag{17}$$

where $\bar{y}$ is the mean of ground truth values.

**Cross-Entropy Loss:**   Used in classification tasks such as CN and neighbor atom prediction, cross-entropy quantifies the difference between predicted and true categorical distributions:

$$\mathcal{L}_{\text{CE}} = -\sum_{i=1}^{N} \sum_{c=1}^{C} y_{ic} \log(\hat{p}_{ic}) \tag{18}$$

where $y_{ic}$ is a binary indicator of the true class, and $\hat{p}_{ic}$ is the predicted probability for class $c$.

**Macro-Averaged F-1 Score:** The harmonic mean of precision and recall computed independently for each class, then averaged:

$$F_1^{\text{macro}} = \frac{1}{C} \sum_{c=1}^{C} \frac{2 \cdot \text{Precision}_c \cdot \text{Recall}_c}{\text{Precision}_c + \text{Recall}_c} \tag{19}$$

This metric ensures equal weight is given to each class, regardless of support size, making it suitable for imbalanced datasets.

## A.3 Experiment Results for XANES/EXAFS Predictions

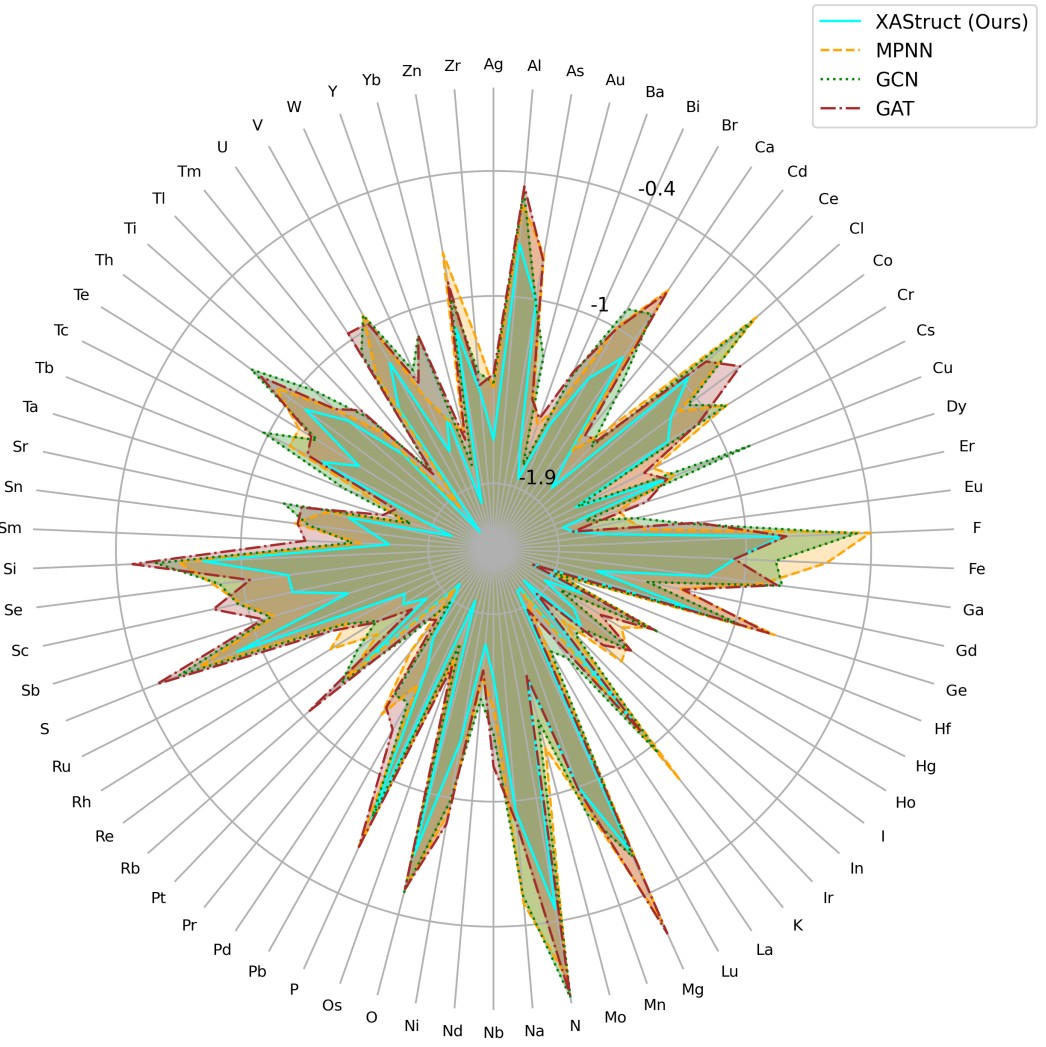

Figure S2: Comparison of K-edge XANES Prediction Model's Performances (Log10 Scale)

**Comparison of Spectral Prediction Errors Across GNN Architectures** Figures S2, S3, and S4 present radar plots comparing the element-wise mean absolute errors ($\log_{10}$ scale) of K-edge XANES, L-edge XANES, and K-edge EXAFS predictions, respectively. These plots evaluate four GNN architectures: our proposed XAStruct, Message Passing Neural Networks (MPNN), Graph Convolutional Networks (GCN), and Graph Attention Networks (GAT).

In all cases, XAStruct achieves consistently lower errors across a broad range of elements, particularly for complex transition metals and heavy elements where coordination environments and

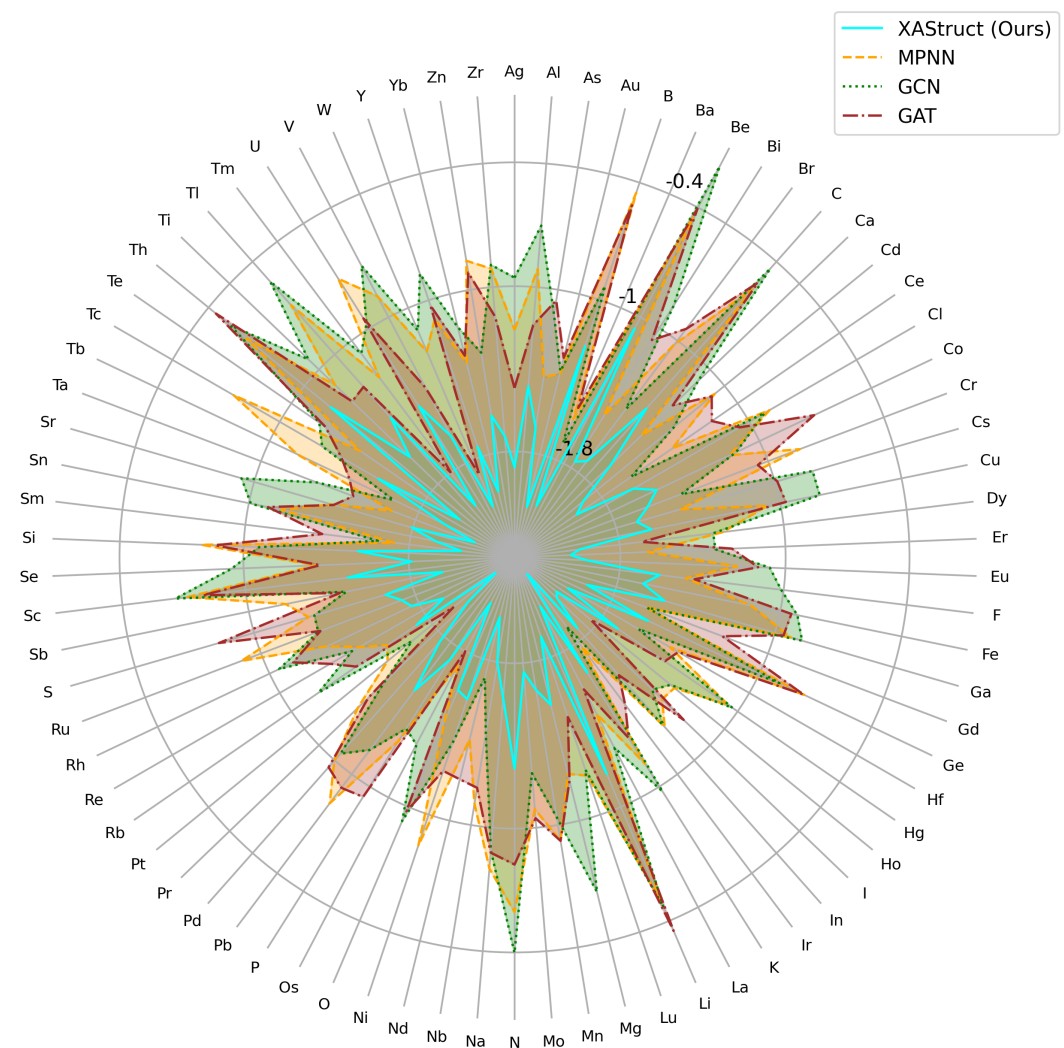

Figure S3: Comparison of K-edge EXAFS Prediction Model's Performances (Log10 Scale)

spectral features become more intricate. The improvement is especially prominent in the L-edge XANES region (Figure S4), where attention-based or convolutional models tend to struggle with fine-edge features.

These results underscore the benefit of XAStruct's architecture in capturing nuanced structure-spectrum relationships, thanks to its domain-informed input encoding and comprehensive learning framework. Nonetheless, the variations in prediction accuracy across elements also suggest space for further optimization, particularly in handling underrepresented edge types or elements with sparse training data.

**Example Results for XANES/EXAFS Predictions**  Figure S7 shows representative comparisons between XAStruct-predicted XAS spectra and ground-truth references for both XANES (top) and EXAFS (bottom) regions. Each subplot overlays predicted (blue) and true (orange) absorption coefficients $\mu(E)$ across energy $E$ for different elements and structural environments.

In the XANES regime (Figure S5), XAStruct captures the global spectral trends and major absorption edge positions, demonstrating strong performance in modeling local electronic structure. However, some limitations are evident: secondary peaks may be under- or over-predicted, and in some cases, sharp features such as white-line or pre-edge signals are smoothed out or shifted. Additionally, predicted curves occasionally lack continuity or local smoothness, particularly in regions

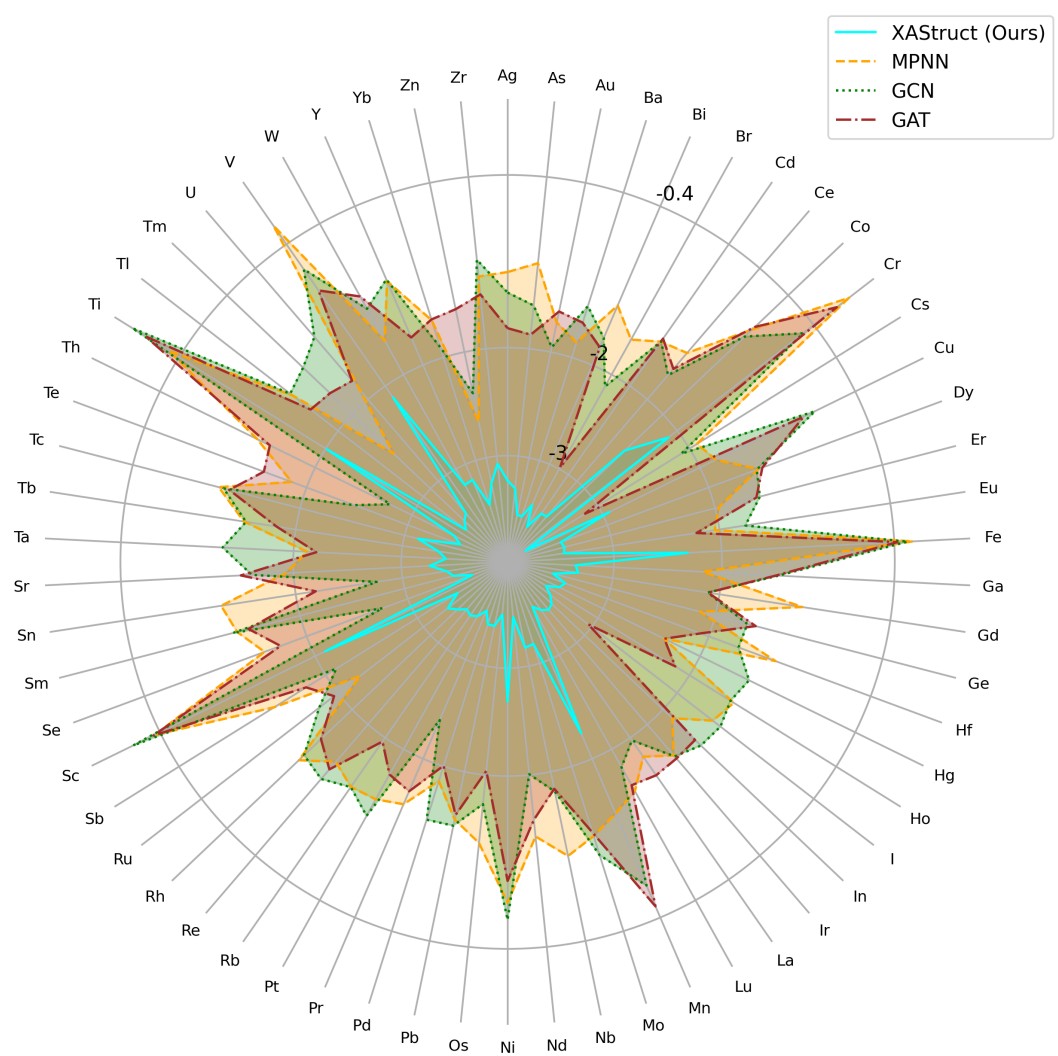

Figure S4: Comparison of L-edge XANES Prediction Model's Performances (Log10 Scale)

of rapid spectral variation, indicating that the model may struggle to fully learn fine-grained electron transition behavior.

In the EXAFS region (Figure S6), the model generally reproduces the oscillatory structure of the signal, including the correct frequency and decay envelope across a wide $k$-range. This suggests effective encoding of photoelectron scattering behavior. Nonetheless, subtle amplitude mismatches, damping discrepancies, and slight phase offsets remain visible in some predictions. These limitations become more pronounced at lower $E$-values, where precise modeling of multiple scattering and longer-range order becomes increasingly difficult.

While these results underscore XAStruct's ability to learn rich structure-spectrum relationships, they also reveal opportunities for improvement. In particular, better peak alignment, improved continuity, and enhanced spectral sharpness will be essential for achieving truly high-fidelity XAS reconstructions.

A.4   EXPERIMENT RESULTS FOR STRUCTURE DESCRIPTOR PREDICTIONS

**MNND Prediction Performance Across Elements and Distributions.**   Figures S8 and S9 evaluate the performance of XAStruct and a baseline RF model in predicting MNND from K-edge XAFS spectra. Figure S8 presents the mean absolute errors (MAEs) of elements on the $\log_{10}$ scale, while

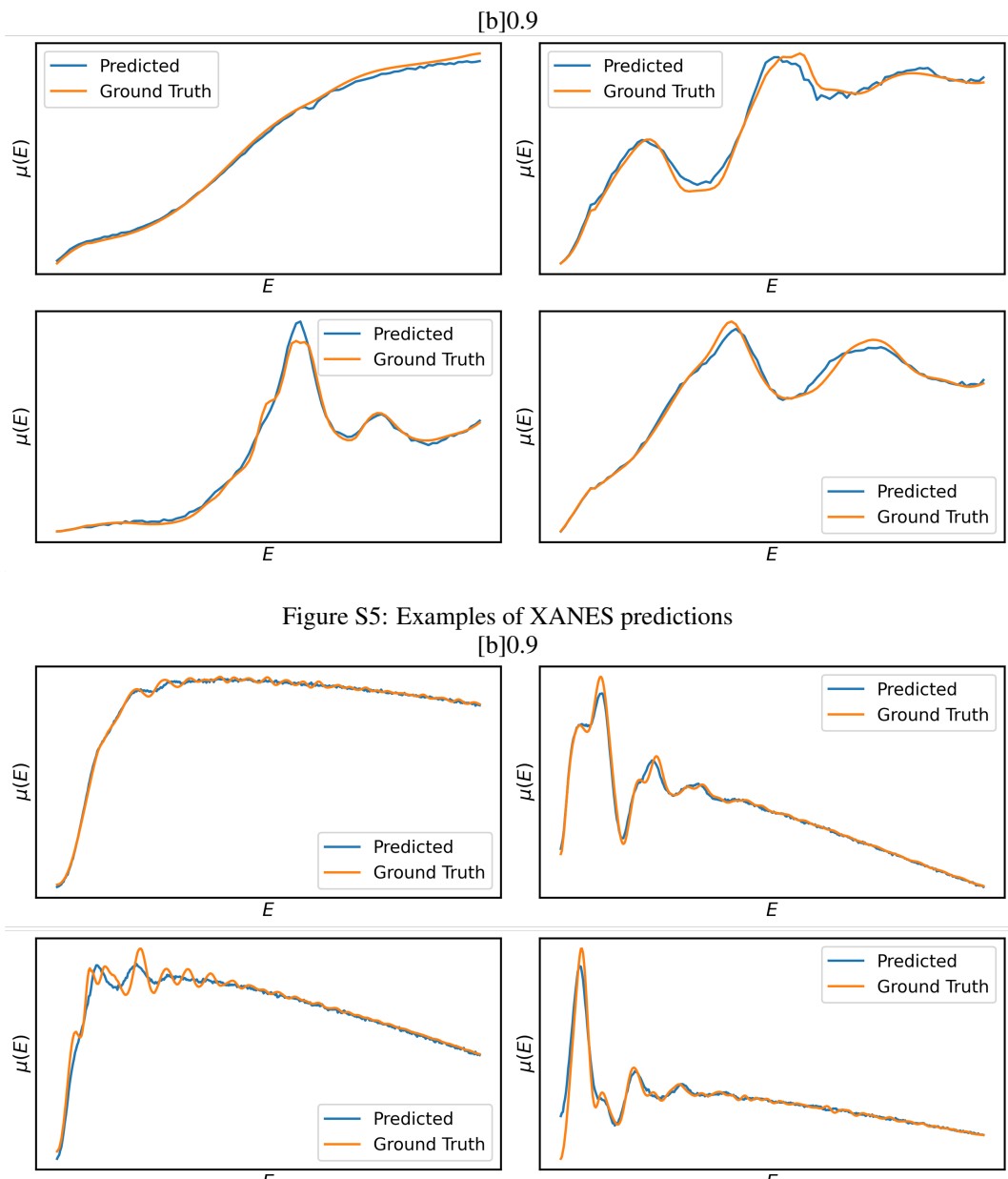

Figure S5: Examples of XANES predictions

Figure S6: Examples of EXAFS predictions

Figure S7: Example results of XAS predictions

Figure S9 shows a correlation scatter plot between the predicted and true MNND values in the dataset.

From Figure S8, XAStruct consistently outperforms the RF model across most elements, achieving lower prediction errors in both common and challenging atomic environments. The reduction in MAE is particularly notable for transition metals and heavier elements, where complex coordination environments often pose difficulties for tree-based models.

Figure S9 further highlights the regression behavior of both models. XAStruct produces a tight distribution around the identity line with an $R^2$ of $0.985$ and MAE of $0.0350 \mathring{A}$. In contrast, the RF baseline shows increased scatter and a slightly lower $R^2$ of $0.983$ and much larger MAE of $0.0557 \mathring{A}$.

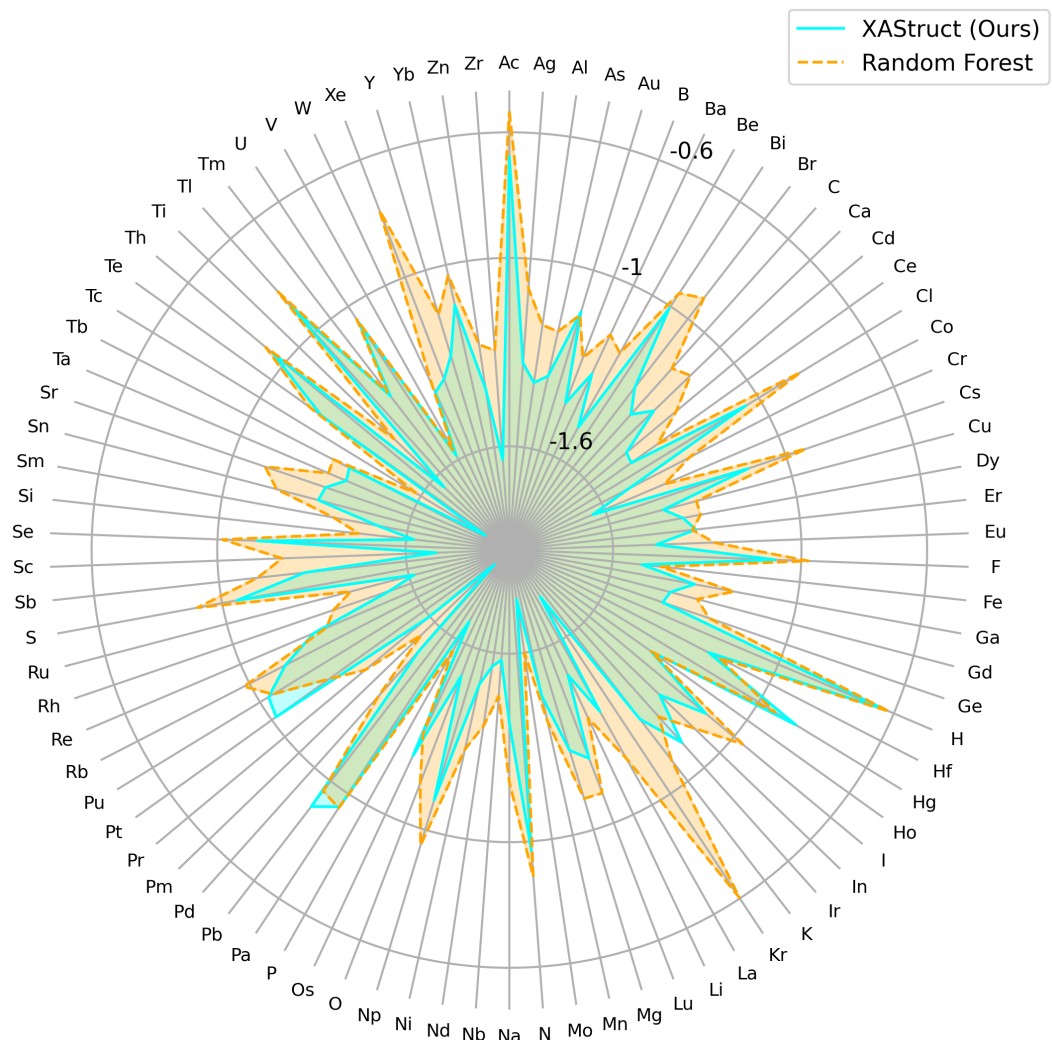

Figure S8: Comparison of K-edge XAFS Based MNND Prediction Model's Performances (Log10 Scale)

As visualized in the figure and quantified in our main text, the RF model exhibits a consistent bias: it underestimates large MNND values and overestimates smaller ones, resulting in a regression slope of 0.92. XAStruct, by contrast, achieves a slope of 0.97, indicating improved fidelity in modeling the true linear relationship.

These results confirm that XAStruct not only generalizes well across elements but also learns smoother, more physically realistic mappings from XAFS spectra to geometric structure descriptors, reducing systematic errors and enhancing model trustworthiness.

**Element-wise Performance Comparison for CN Prediction.**    Figure S10, S11 present radar plots comparing the element-wise prediction performance of two CN classification models: a lightweight random forest (RF, which is a component of the XAStruct framework) and a baseline multi-layer perceptron (MLP). Figure S10 shows the accuracy per element, while Figure S11 shows corresponding F-1 scores.

Overall, the RF model in XAStruct performs slightly better than the MLP baseline across the majority of elements. This is especially evident in cases with more abundant training samples (such as Fe, Cu, etc.), where both models approach high accuracy and F-1 scores. Notably, the RF model

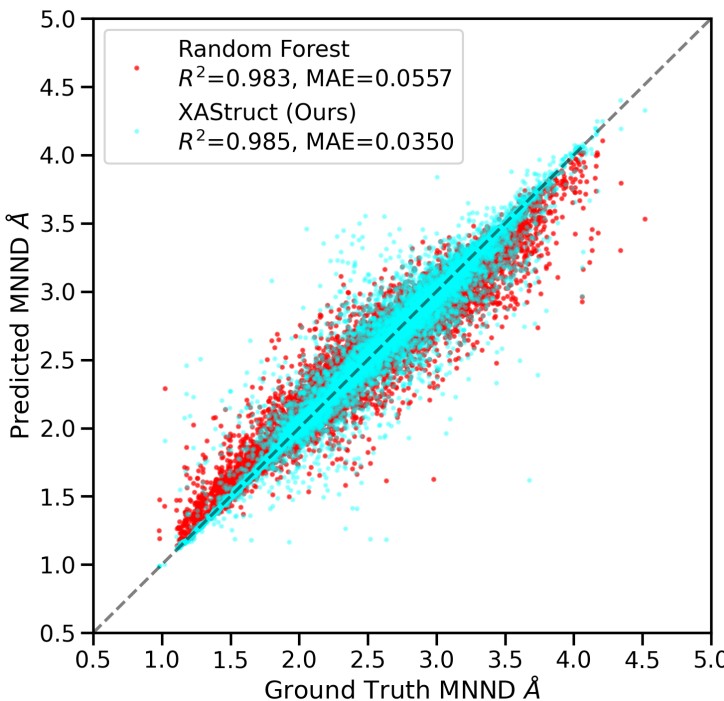

Figure S9: Correlation Plot of MNND Prediction Models

tends to show better stability for rare or underrepresented elements (such as Er, Ho), likely due to its ability to generalize well in low-data regimes without overfitting.

However, both models still struggle with specific elements, especially those with multiple coordination environments or ambiguous bonding geometries (e.g., transition metals and some lanthanides: Tl, Cs, Pb, etc.). Additionally, although radar plots reveal a smooth trend in model behavior across groups of chemically similar elements, localized performance dips highlight potential inconsistencies in featurization or dataset imbalance.

Despite RF's simplicity, its competitive performance and computational efficiency make it a practical choice for CN prediction, particularly in deployment settings where interpretability and speed are favored over architectural complexity.

**Element-wise Performance Comparison for Neighbor Atom Prediction.** Figure S12, S13 compares the performance of the deep learning component of our XAStruct framework and an RF baseline in predicting the identity of neighboring atoms from XAS spectra. Figure S12 shows element-wise prediction accuracies, while Figure S13 displays F-1 scores. This task is framed as a multi-class classification problem where each absorbing atom is assigned a most probable neighbor type.

Overall, XAStruct demonstrates consistent superiority over the RF baseline, especially in terms of F-1 score, indicating improved balance between precision and recall. This advantage is particularly clear for elements with sparse or highly unbalanced training distributions, where RF tends to collapse toward majority classes. XAStruct's graph-aware and element-generalized architecture appears more robust in capturing subtle spectral cues indicative of specific bonding environments.

These trends are supported quantitatively in Table 2, where XAStruct achieves an average accuracy of $92.96_{\pm 0.02}\%$ and F-1 score of $88.76_{\pm 0.01}\%$, outperforming the RF baseline by a large margin (accuracy: **+4.0%**, F-1: **+6.5%**). These results suggest that XAStruct is better suited for high-fidelity spectral interpretation and structure reasoning.

However, some limitations remain for the neighbor atom prediction model. As visible in Figure S13, some elements, particularly those with broad coordination environments or low data representation, still show moderate to low F-1 scores. Additionally, while XAStruct predicts dominant neighbor

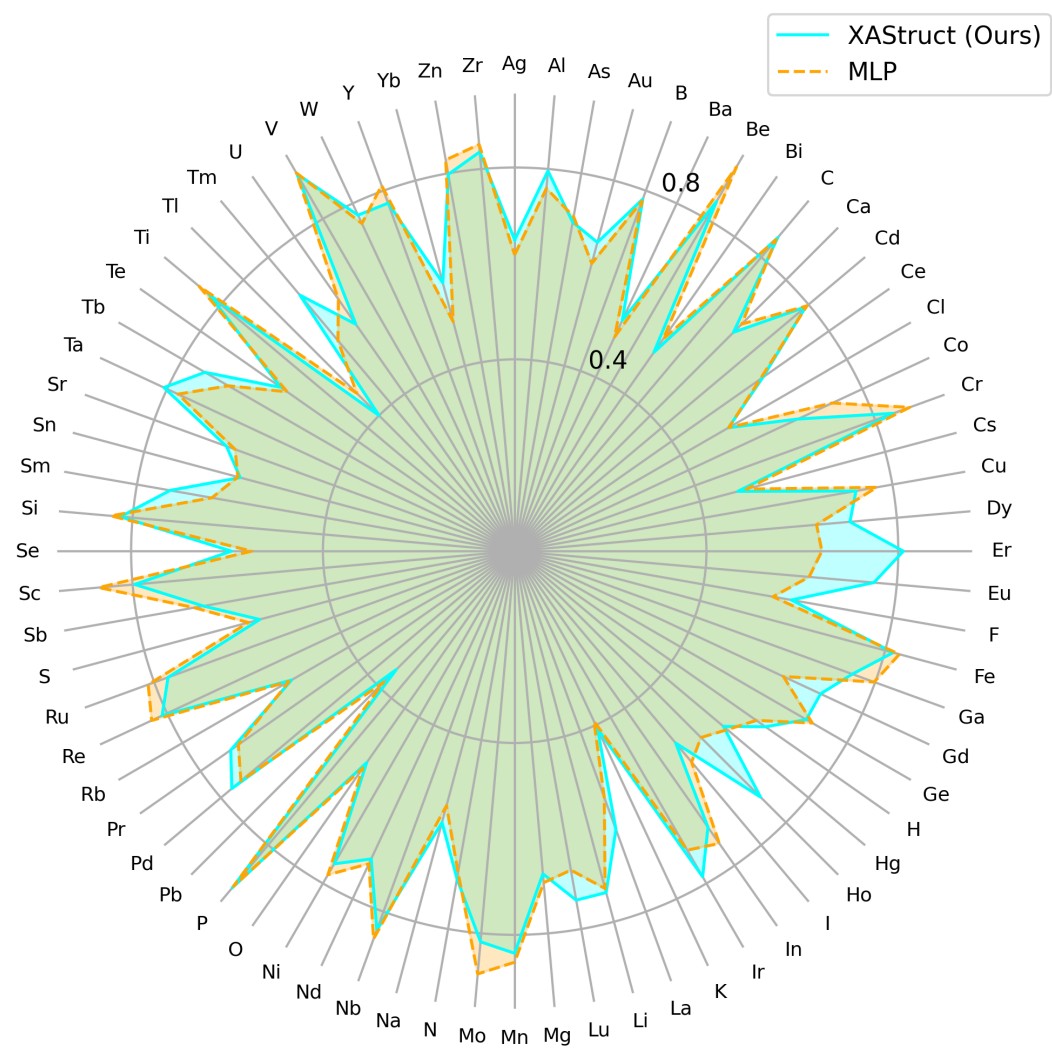

Figure S10: Element-wise Performance Comparison for CN Prediction Top-1 Accuracies

types with high confidence, it does not capture the full neighbor distribution or handle multi-species environments. Future work may benefit from more expressive modeling of neighbor atom uncertainty, attention-based interaction learning, or probabilistic multi-label outputs to handle complex chemical environments.

## A.5   PERIODIC TABLE OF MODEL AVAILABILITY ACROSS ELEMENTS.

As shown in Figure S14, each element block indicates the availability of trained prediction models for the five tasks: XANES spectrum (XA), EXAFS spectrum (EX), coordination number (CN), neighbor atom type (NB), and mean nearest neighbor distance (MNND). Green tags represent successfully trained and evaluated models, while gray indicates missing or underdeveloped support. This coverage highlights dataset and training infrastructure scalability; model scope differs by task (shared vs per-element).

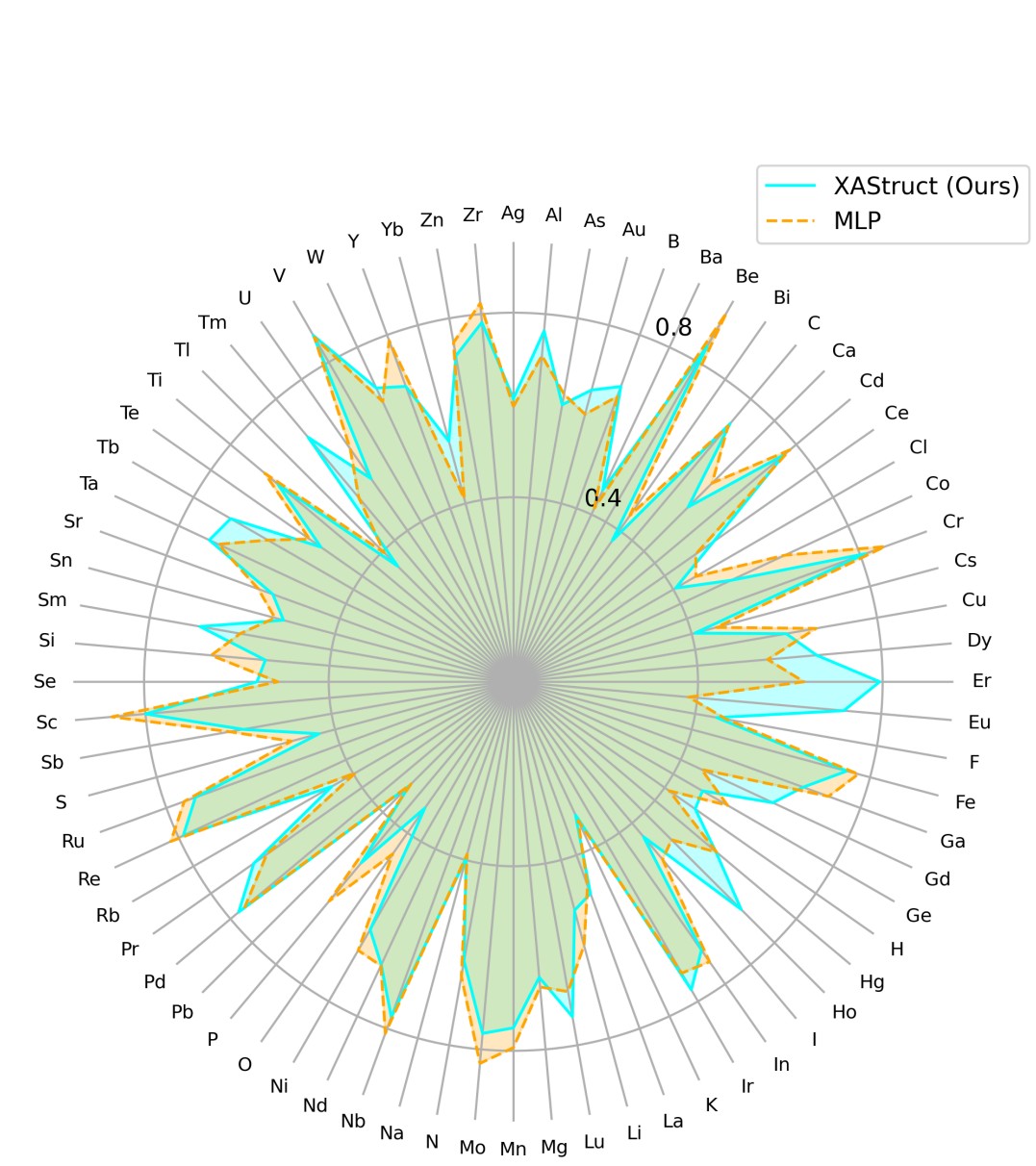

Figure S11: Element-wise Performance Comparison for CN Prediction F-1 Scores.

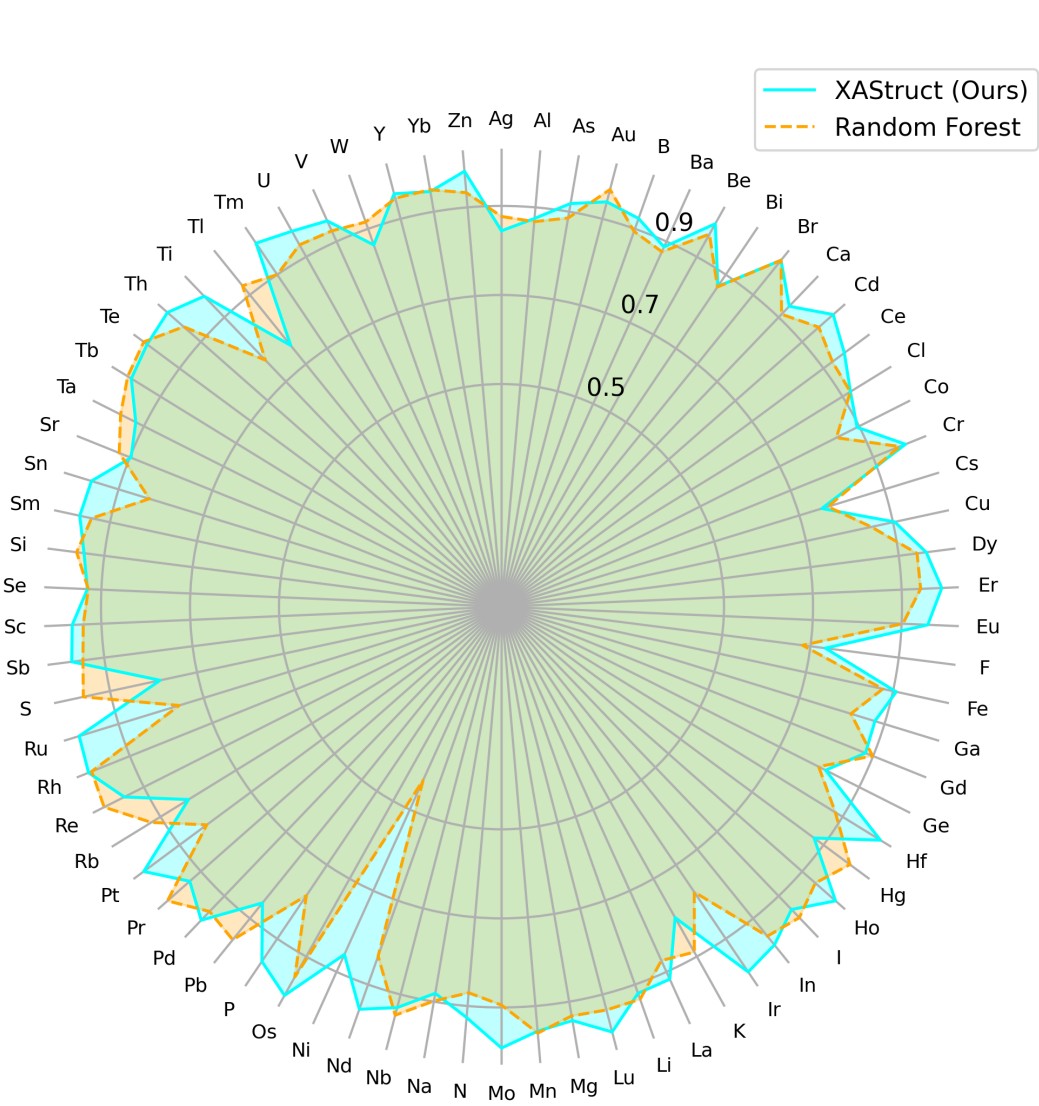

Figure S12: Element-wise Performance Comparison for Neighbor Atom Prediction Top-1 Accuracies.

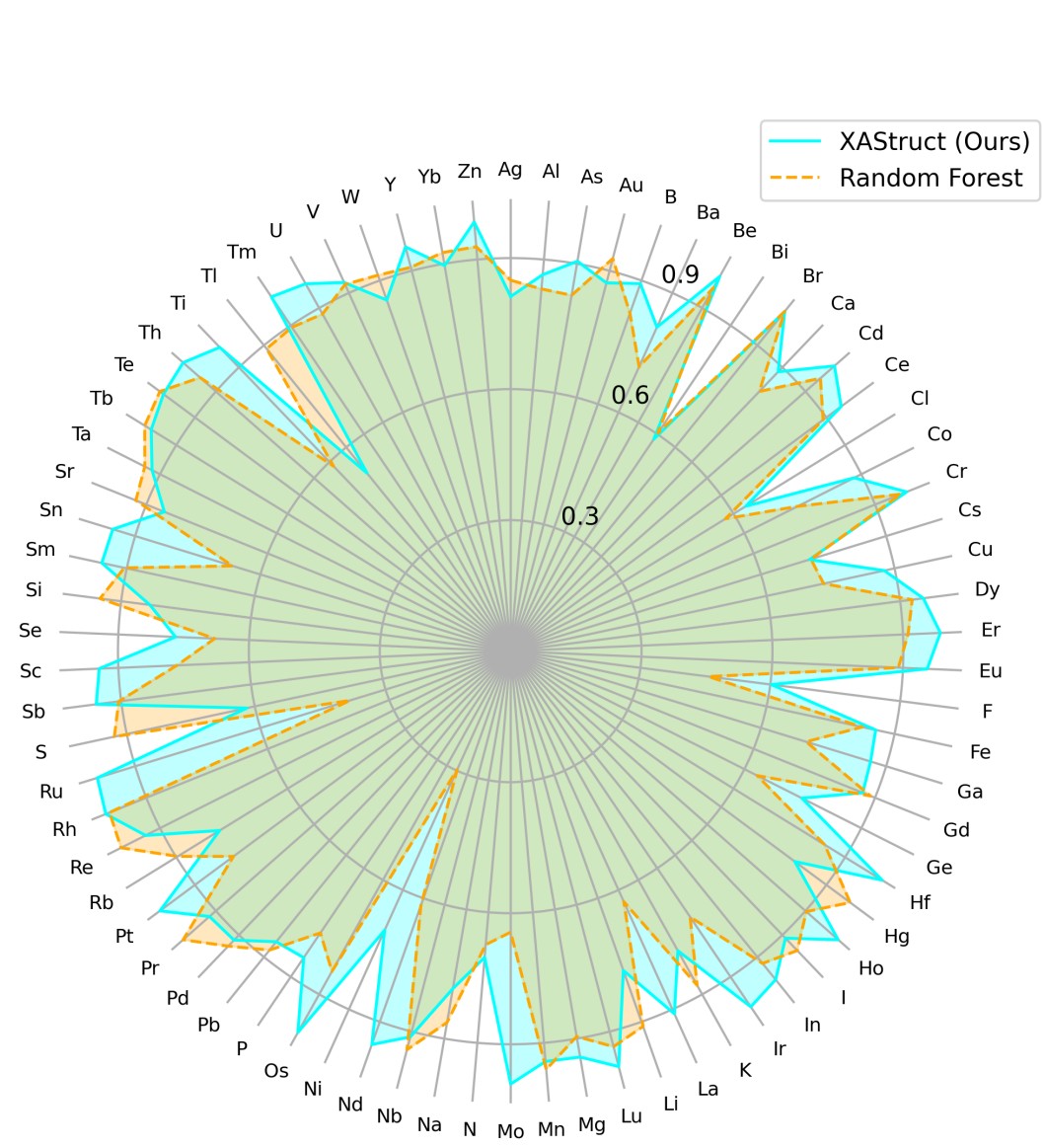

Figure S13: Element-wise Performance Comparison for Neighbor Atom Prediction F-1 Scores.

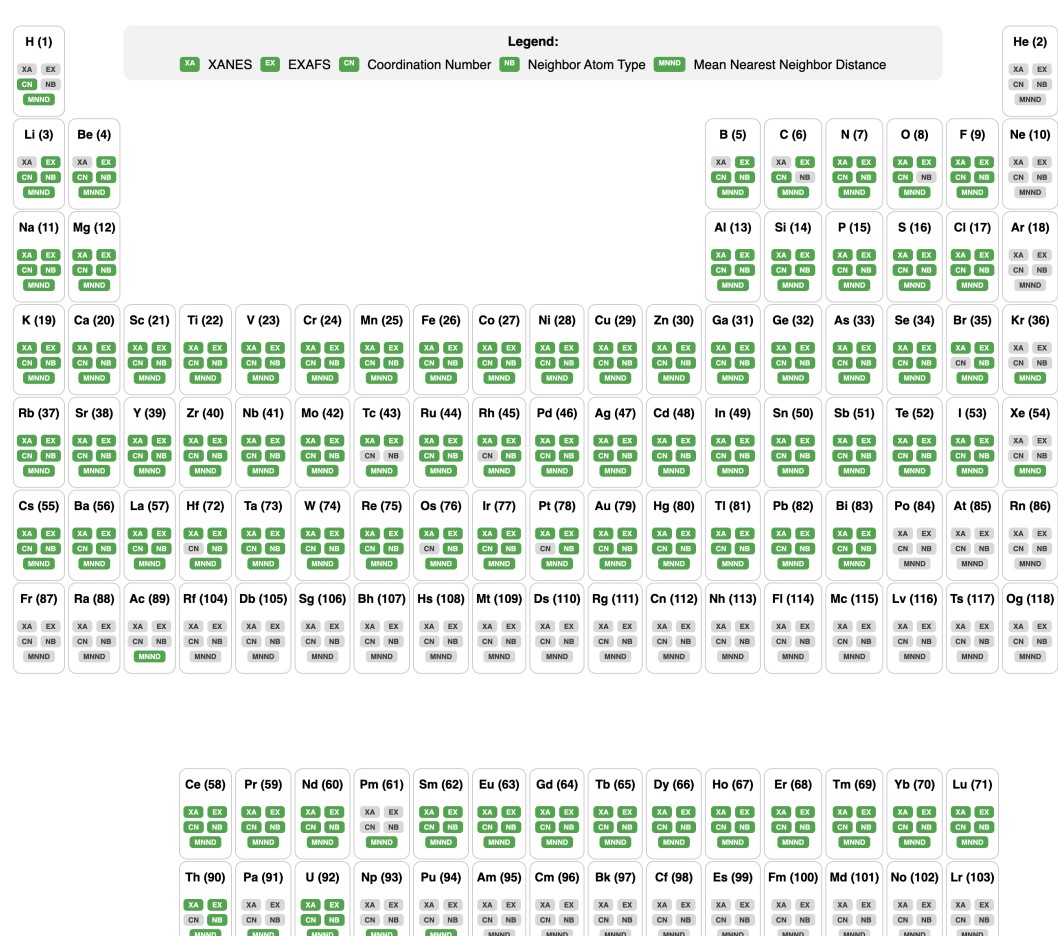

Figure S14: Periodic table of model availability across elements.

