# OpenReview forum: "Inference between Spectra and Structure Across the Periodic Table"
_ICLR.cc/2026/Conference — ICLR 2026 Conference Withdrawn Submission_

### Official Review · Reviewer_itfJ · 2025-10-22

**Soundness:** 2
**Presentation:** 3
**Contribution:** 2
**Rating:** 2
**Confidence:** 5

**Summary:**

This paper proposes a new model, XAStruct, which enables bidirectional mapping between crystal structures and X-ray absorption spectra (XAS). The motivation is sound; however, several key limitations exist, including issues related to data quality and problem definition, as detailed in the weaknesses and questions section.

**Strengths:**

XAStruct is a cross-chemical-system model covering over 70 elements, which is sufficient for practical applications. It can infer key structural information such as coordination numbers, average nearest-neighbor bond distances, and neighboring atom types.

**Weaknesses:**

The key weakness of this paper lies in the problem definition, particularly regarding the mapping from structure to XAS. As the authors introduced in the related physical background, XAS can be divided into two regimes: XANES and EXAFS. XANES mainly reflects multiple-scattering fingerprints, while EXAFS primarily encodes the local atomic configuration.

* Structure-to-XAS mapping: If the structure is given and key information such as local coordination and electronic states is known, there is limited means to fully recover the entire XAS pattern. For generating detailed XAS patterns for experimental matching, the current model does not account for atomic disorder or thermal vibrations, which also significantly affect XAS fluctuations.
* Inverse problem: While the inverse mapping is valid in principle, there are key limitations. First, high-fidelity XAS simulations are currently not feasible because fine structures are related to charge density. Therefore, it is unclear whether the current model can be reliably applied to real patterns without providing any experimental validation.
* Cross-chemical coverage: To cover such a broad chemical space, certain crystals may have elements whose K and L edges are very close, which can hinder the resolution of EXAFS or even XANES signals. The current work does not discuss such scenarios.

**Questions:**

+	How does normalization affect energy alignment, given that each XAS has a different energy range depending on the element, while the absolute energy reflects the electronic level gap?
+	Are there any defined criteria for segmenting XANES and EXAFS regions? In particular, if the XAS spectrum covers two closely spaced K edges, how should the pattern be segmented into multiple stages and concatenated as in Eq. 9?
+	Could multi-task learning be applied to jointly predict MNND, CN, and neighboring atom types, allowing shared latent variables?
+	The current framework may not guarantee sufficient merging of XANES and EXAFS information. Could alternative mechanisms, such as co-learning, be applied? This should be investigated in ablation studies.
+	Are the predictions of MNND and CN consistent? For example, is it possible that CN is predicted to be large while MNND is small?
+	Could the authors provide validation using several experimental real XAS patterns?
+	On page 6 : the citation link for the figure is missing. [Original context: see Figure ?? and Figure S14 for details]

---

### Official Review · Reviewer_qUVL · 2025-10-28

**Soundness:** 3
**Presentation:** 3
**Contribution:** 2
**Rating:** 4
**Confidence:** 2

**Summary:**

This paper introduces XAStruct, a two-pipeline machine learning framework that can both predict X-ray Absorption Spectra (XAS) from crystal structures and infer local structural information directly from XAS input. Trained on a large dataset covering 70 elements, the model generalizes to diverse chemistries and bonding environments. It offers a scalable, data-driven solution for XAS interpretation and local structure inference.

**Strengths:**

By leveraging a large-scale dataset spanning more than 70 elements across the periodic table, XAStruct achieves strong generalization and avoids the element-specific limitations seen in prior models.

**Weaknesses:**

1.Given that XAStruct supports multiple tasks, it would be valuable to explore whether a multi-task learning framework could further enhance performance through shared structural representations.
2.There is a missing figure citation in the fourth line of the Experiment section, which should be corrected for clarity.
3.The design of the SGMLP module appears rather straightforward. It would strengthen the contribution if the authors could provide more insight into the motivation behind this architecture choice and clarify why this specific design is effective for the task.

**Questions:**

1.Since XAStruct supports multiple tasks, would incorporating a multi-task learning framework further enhance performance by leveraging shared structural representations?
2.The SGMLP module design seems relatively straightforward — could the authors elaborate on the motivation behind this architecture and provide evidence or intuition for why this specific design works effectively for the task?

---

### Official Review · Reviewer_MEWh · 2025-10-29

**Soundness:** 2
**Presentation:** 2
**Contribution:** 2
**Rating:** 2
**Confidence:** 4

**Summary:**

This paper addresses the problem of X-ray Absorption Spectroscopy (XAS) by formulating it as a machine learning task. The authors curate two separate models for the forward and backward (inverse) mapping problems between structure and spectra. In addition, the paper provides theoretical background on XAS and discusses remedies for current challenges. The results demonstrate the superiority of the proposed models for both the forward and inverse tasks.

**Strengths:**

This paper highlights the importance of developing machine learning approaches for XAS, which remains a relatively underexplored domain within the AI4Science community.

**Weaknesses:**

For the forward task, the proposed method does not appear to be specifically tailored for XAS. The only additional module introduced in XAStruct is the atom-wise mask, but no ablation study is provided to verify its effectiveness in isolating the absorbing atom and its neighboring atoms. Furthermore, several strong baseline models capable of predicting spectra from inorganic crystal structures (e.g., E3NN, DOSTransformer, ALIGNN, Matformer) are missing. More comprehensive comparisons are needed.

For the inverse model, SGMLP seems to be the only newly added component in XAStruct. However, it is unclear why SGMLP improves performance for the spectra-to-structure task. Does simply increasing MLP complexity lead to better results? A stronger justification of the chosen method is required.

Although the paper claims that the forward and inverse tasks are complementary, the proposed approaches are modeled independently. As a reviewer, I expected a unified or technically integrated framework that bridges the two tasks to achieve superior performance or address the stated challenges. In its current form, the paper lacks clear technical contributions or evidence supporting the necessity of maintaining two separate models.


E3NN: Chen, Zhantao, et al. "Direct prediction of phonon density of states with Euclidean neural networks." Advanced Science 8.12 (2021): 2004214.
ALIGNN: Kaundinya, Prathik R., Kamal Choudhary, and Surya R. Kalidindi. "Prediction of the electron density of states for crystalline compounds with Atomistic Line Graph Neural Networks (ALIGNN)." Jom 74.4 (2022): 1395-1405.
Matformer: Yan, Keqiang, et al. "Periodic graph transformers for crystal material property prediction." Advances in Neural Information Processing Systems 35 (2022): 15066-15080.
DOSTransformer: Lee, Namkyeong, et al. "Density of states prediction of crystalline materials via prompt-guided multi-modal transformer." Advances in Neural Information Processing Systems 36 (2023): 61678-61698.

**Questions:**

The authors claim that the main contribution lies in the two-pipeline architecture, which enables complementary predictions from both structural and spectroscopic inputs. However, it remains unclear which parts of the models are complementary and how the two pipelines enhance each other’s performance

---

### Official Review · Reviewer_29nj · 2025-11-03

**Soundness:** 3
**Presentation:** 4
**Contribution:** 3
**Rating:** 6
**Confidence:** 2

**Summary:**

This paper presents XAStruct, an ML platform to enhance accuracy and completeness of X-ray Absorption Spectroscopy (XAS) predictions. The method has 2 stages:

1. using a GNN encoder to extract features

2. using various classificsation and regression decoders for  spectral prediction, predicting number of neighboring atoms around the absorbing atom, and mean nearest neighbor distance.

**Strengths:**

The paper is well written and motivated. Section 3.1 gives a nice overview of the XAS domain for an ML audience, and section 3.2 does the same for the structural descriptors. The main novelty consists of the GNN, which the authors term “physics principle-aware”., which is largely based on CHGNet. Table 1 reports very strong results against other models for XANES and EXAFS prediction, and table 2 shows strong results for structure descriptor prediction. The ablation study further motivated empirically that the architectural components are well chosen.

**Weaknesses:**

My main concern for this paper is the lack of code. While it is not mandatory, not sharing the code severely limits my ability to assess the veracity of the claims being made. I am also concerned that the paper uses a weak baseline, While this is addressed explicitly (line 402), I remain not fully convinced.

**Questions:**

Please provide further justification for the baselines chosen, and why other baselines were omitted.

---

### Note · Authors · 2026-01-15

I have read and agree with the venue's withdrawal policy on behalf of myself and my co-authors.